# Rapid prototyping and design of cybergenetic single-cell controllers

Sant Kumar[1], Marc Rullan[1] & Mustafa Khammash [1✉]

The design and implementation of synthetic circuits that operate robustly in the cellular context is fundamental for the advancement of synthetic biology. However, their practical implementation presents challenges due to low predictability of synthetic circuit design and time-intensive troubleshooting. Here, we present the Cyberloop, a testing framework to accelerate the design process and implementation of biomolecular controllers. Cellular fluorescence measurements are sent in real-time to a computer simulating candidate stochastic controllers, which in turn compute the control inputs and feed them back to the controlled cells via light stimulation. Applying this framework to yeast cells engineered with optogenetic tools, we examine and characterize different biomolecular controllers, test the impact of non-ideal circuit behaviors such as dilution on their operation, and qualitatively demonstrate improvements in controller function with certain network modifications. From this analysis, we derive conditions for desirable biomolecular controller performance, thereby avoiding pitfalls during its biological implementation.

[1] Department of Biosystems Science and Engineering (D-BSSE), ETH Zürich, Mattenstrasse 26, 4058 Basel, Switzerland. ✉email: mustafa.khammash@bsse.ethz.ch

Propelled by advancements in DNA synthesis, laboratory automation, and a growing repository of characterized biological parts, synthetic biology is starting to bear fruit[1–7]. Recent years have seen a surge of synthetic circuits applied to biotechnology[8] and medicine[9]. However, synthetic circuit construction still presents serious challenges due to unanticipated cross-talk between parts, loading and burden effects, operation in a cellular environment which is inherently stochastic, and long design-cycle time periods, among other reasons. Furthermore, synthetic circuits are usually developed in a context different to that of their end-application, and the circuit's transplantation to new environments comes with complications: changes in culture conditions or host can significantly degrade circuit performance[10].

There are different strategies to mitigate the effect of host and environment context. Circuit reliability can be improved by means of network architecture[11]. A more general approach to engineer robustness is to control the circuit's critical components with feedback regulation, a strategy commonly used in endogenous biological systems[12–14]. The properties of feedback regulation have previously been exploited in synthetic circuits to increase bioprocess yields[15–17], or circuit robustness[18]. However, it is worth noting that previous implementations of in vivo feedback regulation were not capable of perfect adaptation, i.e., convergence to a constant activity level regardless of external disturbances. To achieve this feature, integral action is required[19].

Theoretical implementations of integral controllers solely employing chemical reactions (biomolecular controllers) have been proposed[20–26], but experimental demonstrations remain challenging, with only a few examples of integral (or quasi-integral) implementations (in vivo implementations in[27–30]; in vitro implementation in[31]). The translation of circuit specifications to biomolecular realizations is non-trivial and depends on component availability, characterization, and cross-talk, among others[10,32]. Mathematical modeling is usually employed to identify and alleviate these issues. However, when the target system to be controlled is not quantitatively defined, this approach can lead to large discrepancies between predictions and experimental outcomes.

An engineering strategy commonly used to develop complex, real-time embedded controllers is hardware-in-the-loop (HIL), where the controller being designed is interfaced with a realistic simulation of the system it should steer. HIL is widely used in industries where testing and optimizing the embedded controller in its final application setting is infeasible or very expensive, such as in the automotive or aerospace industries. For example, to ensure an airplane rudder functions suitably over the entire flight envelope, the full rudder and its controlling hardware are interfaced in closed-loop with an aerodynamic computer model of the rest of the airplane. In this way, the expected impact of the rudder dynamics on the airplane flight characteristics can be studied easily for a wide-range of flight conditions. In these cases, HIL vastly decreases development time and costs by shortening the design cycle and minimizing the number of test runs with the real system. To fulfill similar functionalities in a biological setting, we envisioned the Cyberloop (Fig. 1a), a hybrid framework to test and optimize synthetic circuits (biomolecular controllers in this work) under realistic conditions. In the Cyberloop, the targeted in vivo biological system is interfaced at the single-cell level with biomolecular controllers implemented in silico, thereby enabling rapid and cost-effective prototyping. Closing the loop with the true biological system instead of simply using simulations has clear advantages in the design process, as no assumptions need to be made regarding the system's structure or parameters.

The hybrid in vivo/in silico interaction in Cyberloop is achieved via fluorescence measurement and optogenetic activation with light under the microscope[33]. Investigating/Measuring cellular behavior via fluorescent proteins (FP) is a well-known and well-established method in the synthetic biology research community with thousands of FPs available now[34,35] for different cell types. Moreover, optogenetics is a well-known biological technique that uses light to influence biological processes. Most notably, it facilitates a unique capability of controlling gene expression with excellent spatial as well as temporal resolution[36]. An interface of fluorescence measurement and optogenetic activation thus makes this Cyberloop framework applicable to different cell types, and empowers aiming at hundreds of cells individually in a parallel fashion.

Using the Cyberloop with a genetically engineered strain of *Saccharomyces cerevisiae* (Fig. 1b), we first show how the behavior of a biomolecular controller (Autocatalytic Integral Control motif[21]) designed in a deterministic setting drastically changes when put into the stochastic cellular context and provide guidelines to reduce such effects. Secondly, we study the Antithetic Integral Control motif[20], which received broad attention due to its robustness and good performance in stochastic settings. One key assumption required for antithetic controllers to show perfect adaptation is the lack of degradation or dilution experienced by the controller molecules, which has been investigated theoretically[28,37–39]. We here show that dilution rates relevant to *S. cerevisiae* physiology do not significantly degrade the performance of this controller for our target biological network. Further, we analyze two extensions of this control motif, which have been proposed and studied mathematically[40,41]. We exhibit their qualitative effect on the closed-loop performance and stability of the network. Through these examples, we show how the Cyberloop provides actionable insights for the implementation of biomolecular controllers in the cellular environment.

## Results

**The Cyberloop.** Previous strategies for in silico regulation of living cells mostly dealt with population-level feedback, where the readout from all cells (or only one cell) was processed and combined before one common input was given to all the cells[42–48]. There are only a handful of studies implementing single-cell level feedback[33,49,50]. Furthermore, in all of these approaches the control architectures used were deterministic, and followed motifs that are well-established in control engineering, such as proportional control, proportional-integral control, or Model Predictive Control (MPC). While adequate for computer control of bioreactors, these approaches are not typically suitable for bio-molecular control strategies that will be implemented in vivo in the complex and stochastic environment of single cells. To address these challenges, we have expanded a previously published tool for in silico optogenetic control of single cells[33] to include biomolecular (stochastic) control motifs. This experimental platform periodically captures microscopy images of cells placed under the microscope and performs automated image analysis for cell segmentation, tracking, and quantification. The quantified readout from each cell is subsequently passed to its own biomolecular controller, updating the propensities of biomolecular reactions dependent on the measured species (Fig. 1a). The reaction network is then simulated with stochastic setting[51] until the next measurement time-point, when the controller output is fed back to the cell in the form of an optogenetic input. In this framework (Supplementary Fig. 4), the controller output (computed light intensity for cell stimulation) is applied after a delay of one sampling time interval which was set to be 2 min in our experiments. For the biological target system (Fig. 1b) used in our experiments, this small delay has negligible impact on the controller performance[33]. The sampling/imaging time period of

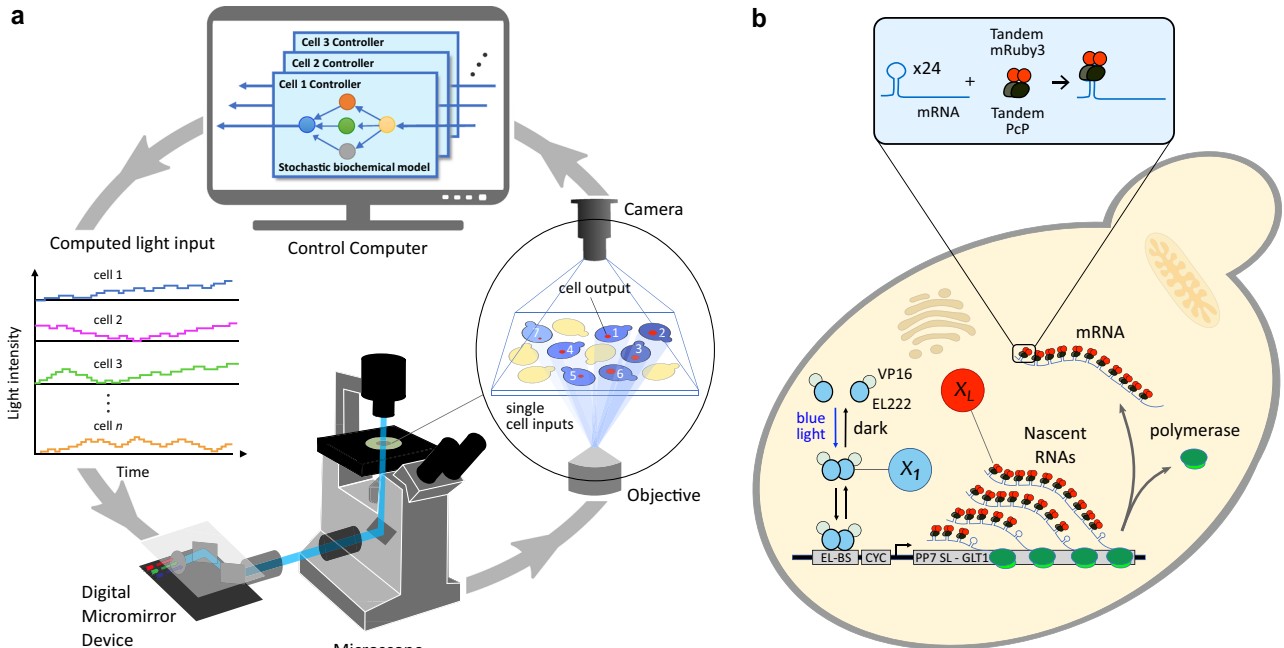

**Fig. 1 Experimental framework for stochastic optogenetic control of single cells. a** The Cyberloop. Light responsive cells are grown in a monolayer at the microscope sample plane and imaged periodically. Image analysis is then performed to obtain the coordinates of each cell, track cells across frames, and quantify cellular readouts of interest. These cellular readouts are used to update the propensities of a stochastic biomolecular controller reaction network simulation. The controller species abundance of such a network is used to determine the amount of light that each cell will receive at the next imaging cycle. Light is administered to the individual cells through a custom-designed light-projection hardware[33] attached to the microscope. **b** Optogenetic induction of gene expression and real-time visualization system in *Saccharomyces cerevisiae*. Blue light promotes homodimerization of EL222-VP16, which is then able to bind its target site EL-bs[62]. Once bound, the optogenetic system promotes gene expression via its activation domain, VP16. Visualization of nascent RNA counts is achieved by means of the tdPCP-PP7 system[52]. EL222-VP16 regulates the expression of an RNA containing 24 stem-loops at its 5' end. These stem-loops are recognized by the PP7-mRuby3 fusion protein. During transcription, a large number of fluorescent proteins localizes in a diffraction-limited spot inside the nucleus. The fluorescence intensity of such a spot is proportional to the number of nascent RNAs[33].

2 min was chosen based on the computation time required to run our software routine (executing cell segmentation, tracking, quantification, and stochastic controller simulation for individual segmented cells) between consecutive sampling in our experiments (Supplementary Fig. 8). Approximately 75 to 100 cells were targeted and tracked for the full-duration of each experiment.

We chose a genetically modified strain of *S. cerevisiae* for our experiments. The biological target variable, placed under control throughout this study, was the nascent RNA counts of tdPCP-PP7 system[52] as shown in Fig. 1b. The real-time visualization/measurement of nascent RNAs was achieved by using a live-cell fluorescent reporter (PP7-mRuby3 fusion protein). Gene expression has been shown to occur in bursts[33] displaying a high degree of variability both among cells and over time. This makes it an ideal system to test the performance of stochastic biomolecular controllers. To facilitate optogenetic activation we used a light-activated transcription factor (Fig. 1b). And we employed the Digital Micromirror Device (DMD) based custom-built projection hardware and software (developed in[33]) to direct light to individual cells, placed under the microscope, with high spatio-temporal precision. The reader is referred to Methods section for further technical details.

**Autocatalytic Integral Controller**. We first used the Cyberloop to characterize the Autocatalytic Integral Controller topology[21], which belongs to a class of biomolecular control architectures implementing integral feedback with one sole controller species[24,25]. This minimal integral network topology (Fig. 2a) guarantees set-point tracking and robust perfect adaptation in deterministic setting, and it has been shown that its metabolic

load on the host cell can be tuned to a minimal level compared to all possible controllers. However, its performance in the presence of stochastic variations remains unexplored to date.

As the first test of this motif's behavior when implemented through stochastic chemical reactions (Fig. 2a, right), we showed its effectiveness for set-point tracking when the copy-number of controller species $V$ is constrained to be strictly positive. Under this condition, this motif displays the desired set-point tracking performance (Fig. 2b). However, when no constraints are placed on the copy-number of $V$ and its time evolution is modeled stochastically, it will eventually (with probability one) reach an abundance of zero molecules and not change further. This is because $V = 0$ is an absorbing state for this system. The only source of $V$ is a positive feedback mechanism, requiring the presence of $V$ itself (Fig. 2a). Therefore, the absence of $V$ implies the impossibility of this autocatalytic reaction to fire, and no more molecules of $V$ can further be produced. We recapitulated this behavior in our experiments (Fig. 2c). We also found that by decreasing the value of gain parameter, $k$, we could extend the average time needed for the network to reach the absorbing state (Fig. 2c, bottom). At steady-state, for $X_L$ to track the set-point a certain amount of $X_1$ will be required. This $X_1$ is produced in the actuation reaction (third reaction in Fig. 2a, right) whose propensity is $kV$. This implies that decreasing the value of $k$ would result in higher abundance of $V$ to maintain the production of the required amount of $X_1$. This can be observed in Fig. 2c (inset). At time $t = 100$ min, the experiment with lower $k$ value exhibits higher $V$ abundance in cells compared to the one with higher value of $k$. And this higher abundance of V, in turn, prolongs the time for the system to reach the absorbing state.

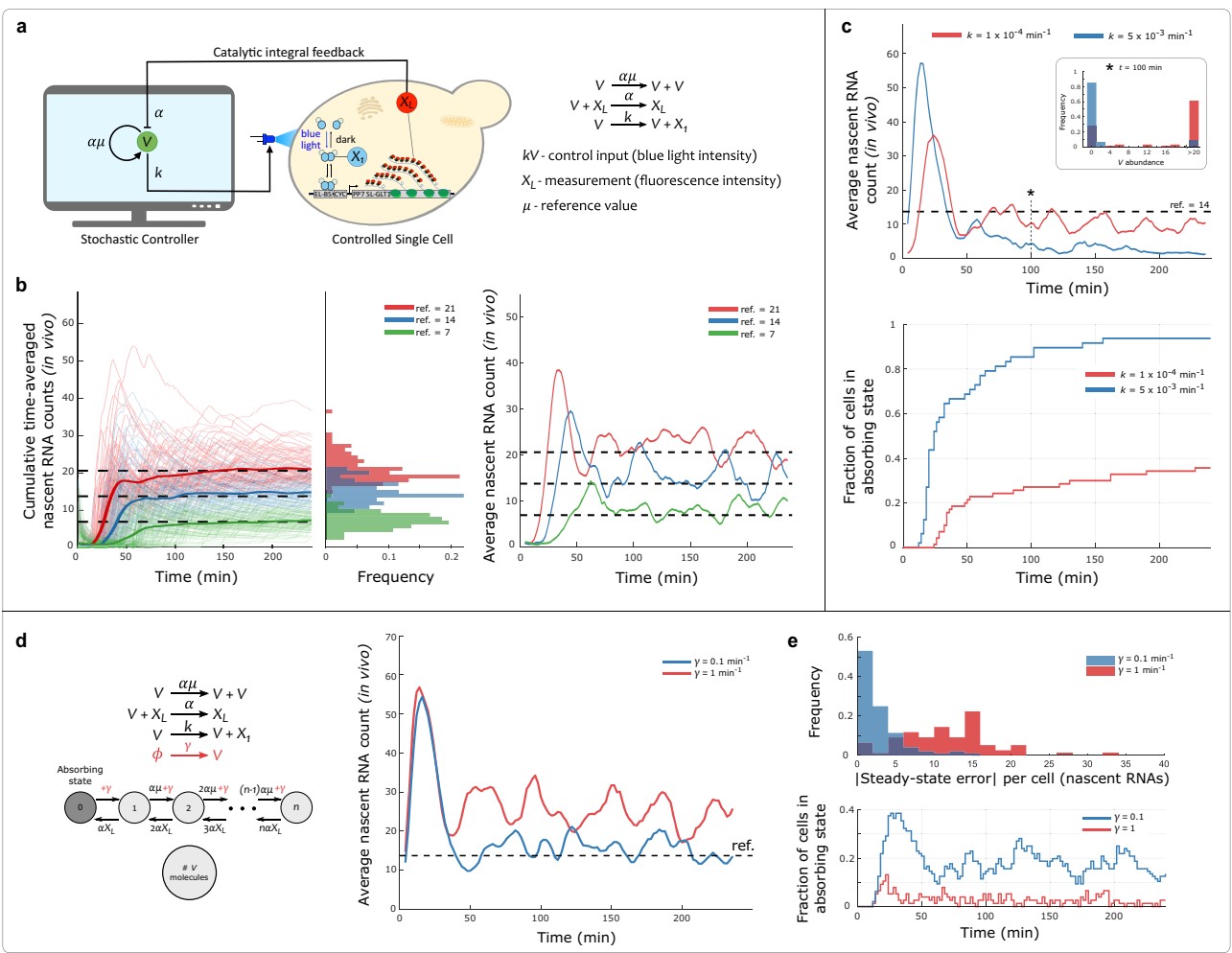

**Fig. 2 Autocatalytic Integral Controller in Cyberloop. a** Cyberloop implementation of the Autocatalytic integral biomolecular controller. Left: the cellular readout, $X_L$, is measured via fluorescence microscopy. Its copy-number, quantified from fluorescence intensity, is used to update the reaction propensities of the controller network, implemented in a computer. The control action is performed at regular intervals by applying a light input to the controlled cell. The applied light input intensity is proportional to the controller species ($V$) copy-number obtained from stochastic simulation in the computer. Right: Autocatalytic integral controller reactions[21]. **b** Demonstration of set-point tracking by the Autocatalytic Integral Controller motif (under the constraint $V > 0$). Left: three Cyberloop runs with different reference values (dashed lines) were performed. Thin lines indicate cumulative time averages of nascent RNA counts in individual cells, while thick lines represent the population average. Center: distribution of the average nascent RNA count per cell over the course of the experiment. Right: population average of nascent RNA count across cells as time progresses in these three experiments (Experimental parameters: $k = 0.005\,\text{min}^{-1}$, $\alpha = 0.01\,\text{min}^{-1}$, initial $V = 1$ and $\mu = (7, 14, 21)$; Number of cells: red - 98, blue - 86, green - 102). **c** Effect of absorbing state on the Autocatalytic Integral Controller motif. Two set-point tracking experiments with differing values of parameter $k$ show the effect of this parameter on the average time needed for the controller species abundance to reach zero, the absorbing state of the system. Top: time-course evolution of average nascent RNA counts for the controller with high and low k values. Inset: distribution of $V$ copy-numbers for all the cells at time $t = 100\,\text{min}$ in the two experiments. Bottom: shows the fraction of cells in the absorbing state as a function of time. As time progresses in the experiment, more cells fall into the absorbing state due to stochasticity, thus making the controller ineffective for set-point tracking under stochastic setting (Experimental parameters: $\alpha = 0.01\,\text{min}^{-1}$, initial $V = 300$ and $\mu = 14$; Number of cells: red - 70, blue - 48). **d**, **e** Effect of promoter leakiness on Autocatalytic Integral Controller – addition of a basal production (leakiness) rate of controller species $V$ eliminates absorbing state. **d** Left: an additional reaction (in red) is added to the Autocatalytic Controller motif. With this addition, the absorbing state is removed, as the basal production rate enables the system to leave the state $V = 0$ with propensity $\gamma$. This basal production rate of $V$ affects the tracking properties of the controller under stochastic setting. Right: time-course evolution of average nascent RNA counts for the controller with two different basal production rates. **e** Bottom: fraction of cells in the absorbing state as time progresses in the two experiments. Top: steady-state error (absolute value) distribution of nascent RNA counts for all the cells. Sufficiently small basal production rate eliminates the absorbing state and makes the controller effective in robust set-point tracking even under the stochastic setting. (Experimental parameters: $k = 0.005\,\text{min}^{-1}$, $\alpha = 0.01\,\text{min}^{-1}$, initial $V = 300$ and $\mu = 14$; Number of cells: red - 76, blue - 96). Source data are provided as a Source Data file.

In biology, components used to build synthetic networks often present behaviors not necessarily present in the idealized design. The Cyberloop allows incorporation of such non-ideal biological effects in the controller network, as for example promoter "leakiness" (Fig. 2d). The addition of a basal expression rate of

our controller species, $V$, modifies the system behavior and eliminates the robustness and perfect tracking properties of our network (Fig. 2d, right in red; Supplementary Information Section 1.1.1). However, experimental results show that it can also have a positive effect if balanced properly with the rest of the

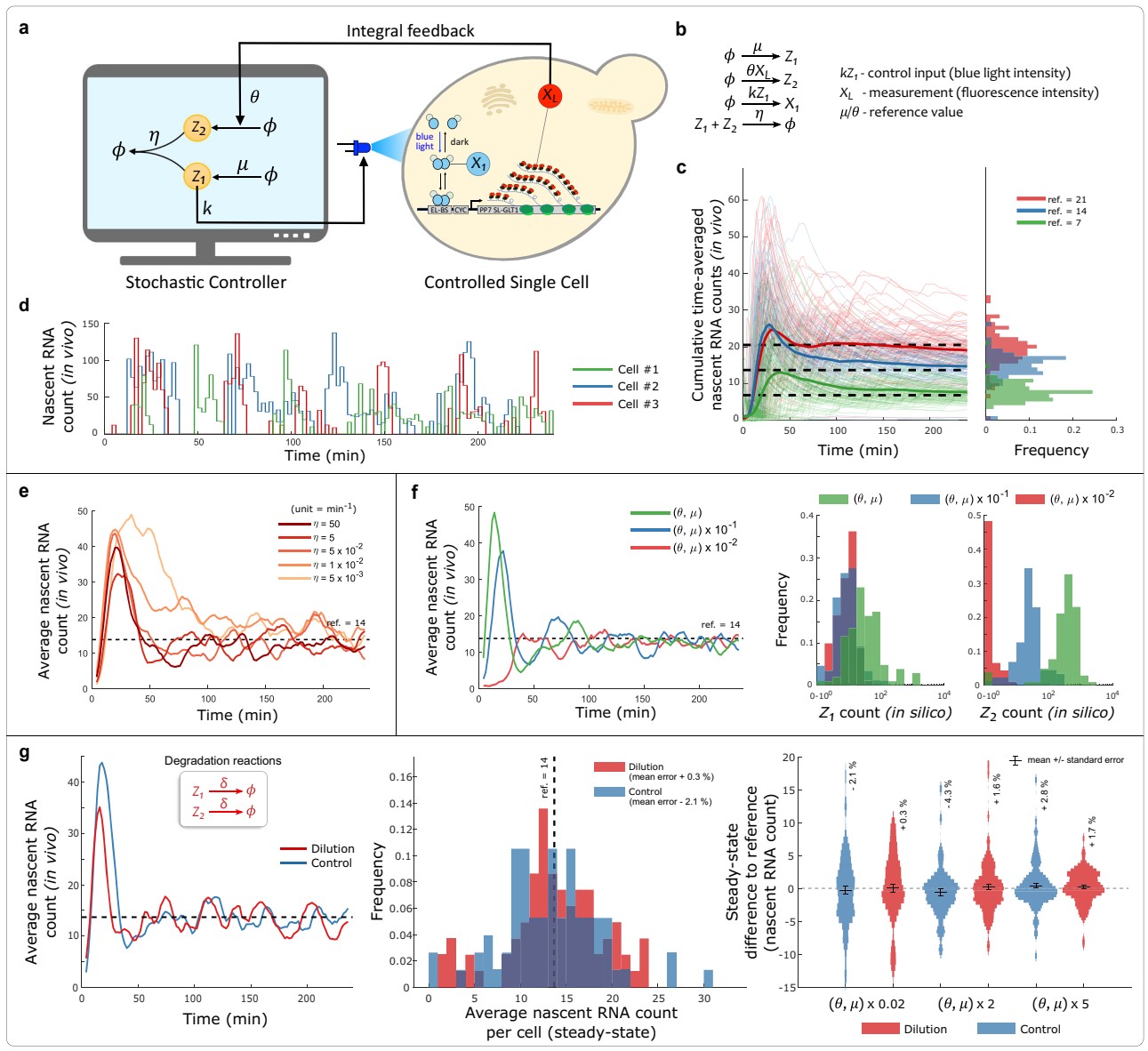

system parameters, as it eliminates the absorbing state (Fig. 2d, left). As can be seen in (Fig. 2d and e, blue), proper tuning of $\gamma$, the basal production rate of $V$, enables long-term set-point tracking without the need for positivity constraint on $V$, and with minimal steady-state error even in a stochastic setting.

**Antithetic Integral Controller.** The Antithetic Integral Controller was the first biomolecular control motif to take into account the stochasticity present in chemical reaction networks, and even draw benefits from it[20]. This controller is composed of two species, $Z_1$ and $Z_2$, which interact with the controlled system and with each other to robustly regulate $X_L$ abundance (Fig. 3a, b). In a similar fashion as the Autocatalytic controller, we implemented the Antithetic controller in the Cyberloop to study its properties when connected to a real biological system.

One key characteristic of integral controllers is their ability for perfect set-point tracking. We first observed that indeed this controller can accurately bring the average nascent RNA count to different desired reference levels (Fig. 3c). In addition to this, we also observed that it can track dynamic references or set-points

during the course of the experiment (Supplementary Fig. 6). Next, we tested its ability to precisely regulate gene expression in the presence of strong stochastic fluctuations present at the level of the biomolecular controller, and at the level of the biological system under control, which represents high intrinsic noise due to low copy number of the involved molecules. For this, we operated the controller circuit in parameter regimes where $Z_1$ and $Z_2$ exist in single-digit abundances, and observed no change in the controller's steady-state tracking performance (Fig. 3f). This supports the theoretical analysis of the control motif, which had shown that stochastic noise was not detrimental to its performance[20]. Next, we focused on analyzing the role of the sequestration reaction (rate $\eta$) between $Z_1$ and $Z_2$ on the closed-loop dynamics of the system. We observed that higher sequestration rates led to improved transient dynamics, e.g. faster settling times. However, we found that values of $\eta$ of the same order of magnitude as the production rates of $Z_1$ and $Z_2$ achieved reasonable performances, and further increases in $\eta$ produced only marginal gains (Fig. 3e). Therefore, $Z_1$ and $Z_2$ candidate pairs do not necessarily require an extraordinarily high binding

**Fig. 3 Antithetic Integral Controller in Cyberloop. a**, **b** Cyberloop implementation of the Antithetic Integral Controller motif, and the associated controller reactions[20]. The cellular readout, $X_L$, is used to update the reaction propensities of the controller network involving species $Z_1$ and $Z_2$. The applied light input (control input) intensity is proportional to the copy-number of controller species $Z_1$, which is obtained from stochastic simulation in the computer. **c** Demonstration of set-point tracking. Left: three Cyberloop runs with different reference values (dashed lines) were performed. Thin lines indicate cumulative time averages of nascent RNA counts in individual cells, while thick lines represent the population average. Right: distribution of the average nascent RNA count per cell over the course of the experiment (Experimental parameters: $k = 0.1\,\mathrm{min}^{-1}$, $\eta = 5\,\mathrm{min}^{-1}$, $\theta = 0.02\,\mathrm{min}^{-1}$ and $\mu = (7, 14, 21) \times \theta\,\mathrm{min}^{-1}$; Number of cells: red - 94, blue - 76, green - 110). **d** Single-cell output traces of three randomly chosen cells in the Cyberloop experiment shown in (**c**) with ref. = 21. **e** Effect of annihilation reaction rate on the closed-loop system dynamics. Five independent experiments with differing annihilation rates $\eta$ were performed. The plot shows the time-course evolution of average nascent RNA counts in those experiments. Lower values of $\eta$ (lighter color) led to longer transients, but did not affect steady-state properties of the system (Experimental parameters: $k = 0.1\,\mathrm{min}^{-1}$, $\theta = 0.02\,\mathrm{min}^{-1}$ and $\mu = 14 \times \theta\,\mathrm{min}^{-1}$; Number of cells: in increasing order of $\eta$ values - 79, 80, 90, 104, 100). **f** Effect of controller species production rates. Three independent set-point tracking experiments with a common reference (dashed line), but differing values of $\theta$ and $\mu$ were performed. Left: time-course evolution of average nascent RNA counts in those experiments. Higher $\theta$ and $\mu$ values led to larger overshoot and longer settling times. Right: steady-state distribution of $Z_1$ and $Z_2$ copy-numbers over all the cells. Lower $\theta$ and $\mu$ values resulted in lower copy-numbers of $Z_1$ and $Z_2$ at steady-state. The Antithetic Integral Controller operates well, even in very low (<10) copy-number regime of controller species (Experimental parameters: $k = 0.01\,\mathrm{min}^{-1}$, $\eta = 5\,\mathrm{min}^{-1}$, $\theta = 2\,\mathrm{min}^{-1}$ and $\mu = 28\,\mathrm{min}^{-1}$; Number of cells: red - 91, blue - 87, green - 104). **g** Effect of physiological dilution on the operation of Antithetic Control motif - addition of degradation reactions (inset) for the controller species of the Antithetic Controller. The degradation rate $\delta$ was set as $\frac{\log(2)}{\text{Doubling Time}}$, that is, the dilution rate of the intended host cell type (*S. cerevisiae*). In these Cyberloop experiments, this dilution rate was found to have negligible impact on the controller performance. Left: shows the averaged population response over time of two experiments with (red) or without (blue) degradation reactions (Experimental parameters: $k = 0.1\,\mathrm{min}^{-1}$, $\eta = 5\,\mathrm{min}^{-1}$, $\theta = 0.02\,\mathrm{min}^{-1}$ and $\mu = 14 \times \theta$; Number of cells: red - 81, blue - 76). Center: contains the histogram of average nascent RNA counts per cell at steady-state for the same experiments shown in (Left). Right: contains the distribution of steady-state errors for three parameter combinations. Inset percentage values are the percentage mean errors from the reference, and the error bars denote mean $+/-$ standard error. (Experimental parameters: $k = 0.1\,\mathrm{min}^{-1}$, $\eta = 5\,\mathrm{min}^{-1}$, $\theta = 1\,\mathrm{min}^{-1}$ and $\mu = 14\,\mathrm{min}^{-1}$; Number of cells: from left to right in the violin plot - 76, 81, 67, 93, 97, 114). Source data are provided as a Source Data file.

affinity to warrant a good controller performance for the given biological system.

A key assumption required for the Antithetic controller to display perfect disturbance rejection and tracking is that the degradation of controller species $Z_1$ and $Z_2$ is negligible. This assumption is challenged in fast-dividing cells, or when $Z_1$ and $Z_2$ present a short half-life. The consequences of degrading controller species are that the integral term, or the memory in the system, is gradually lost, as explained in[37]. A theoretical treatment of the issue shows that degradation introduces an error on the steady-state tracking properties of the system (Supplementary Information Section 1.2.1), which is dependent on the controller's parameters. To test out the practical implications of dilution due to cell growth in *S. cerevisiae*, the organism where the controllers are supposed to be embedded, we introduced degradation terms in $Z_1$ and $Z_2$ (Fig. 3g, inset) of the order of yeast's cell cycle period. The cell cycle period or the doubling time of our engineered yeast strain was found to have a distribution closely resembling a Poisson distribution with a mean doubling time of 77 min (Supplementary Fig. 7). During the experiment, doubling times (for computing dilution rates, $\delta = \frac{\log(2)}{\text{Doubling Time}}$) of individual cells used in their in silico controller network were randomly sampled from this distribution. We found that for the given biological system network (Fig. 1b), dilution does not play a significant role in the steady-state tracking properties of the controller resulting in negligible tracking error over a broad range of controller parametrizations (Fig. 3g, right).

**Antithetic Integral Rein Controller.** We next implemented the "Rein" extension[40] of the Antithetic control motif in the Cyberloop and observed its implications on the closed-loop system behavior. In addition to all biomolecular reactions in the Antithetic controller network, this motif includes an extra direct negative feedback from the controller species $Z_2$ to the control target species $X_L$ represented by the reaction highlighted in (Fig. 4a in red). As investigated in[40], considering a classical gene-expression model as the system to be

controlled, this additional repressing feedback from $Z_2$ and the normal actuating influence from $Z_1$ have an opposing effect on the system output which in turn results in improved system dynamics. In the Cyberloop, we only have two-way communication (output measurement through fluorescence microscopy and actuation though light stimulation) channel between the in silico controller simulation and the real biological system but this extended motif requires one more channel to implement the additional reaction. To circumvent this problem we added an in silico molecule $Y$, the new symbolic system output to be controlled. This molecule is actuated by the real biological system output $X_L$ (nascent RNA count). Thus, the target system network to be controlled was a hybrid network with real biological entities $X_1, \ldots, X_L$ and in silico species $Y$ (Fig. 4b). This allowed us to implement and carry out experiments with this motif in the Cyberloop with $Y$ as our output variable of interest.

In these experiments, we observed that this controller motif achieves the desired set-point tracking by bringing the output molecule $Y$ count to different set-points (Fig. 4c). Further, we found that this Antithetic Integral Rein Controller indeed improves the system response characteristics, achieving lower overshoot, faster settling time, and lower variance compared to the original Antithetic control motif (Fig. 4d, top and bottom). This observation is in line with the theoretical findings presented in[40]. We also observed that for the given parameter values, the "rein" extension requires very low abundance of controller species $Z_2$ to achieve set-point tracking when compared to the Antithetic control motif (Fig. 4d, center). Furthermore, we found that when the output degradation rate ($\delta$ in Fig. 4b) is high enough, this extended Antithetic motif doesn't lead to significant qualitative improvement in controller tracking performance over the original Antithetic motif (Fig. 4e). This suggests that under such conditions when the system output has a considerable degradation rate of its own, the addition of this extra negative "rein" feedback to the output within an Antithetic motif may not be warranted. We also briefly studied the favorable performance of this controller motif in the presence of a basal production reaction of the output molecule. The results are shown in Supplementary Fig. 9.

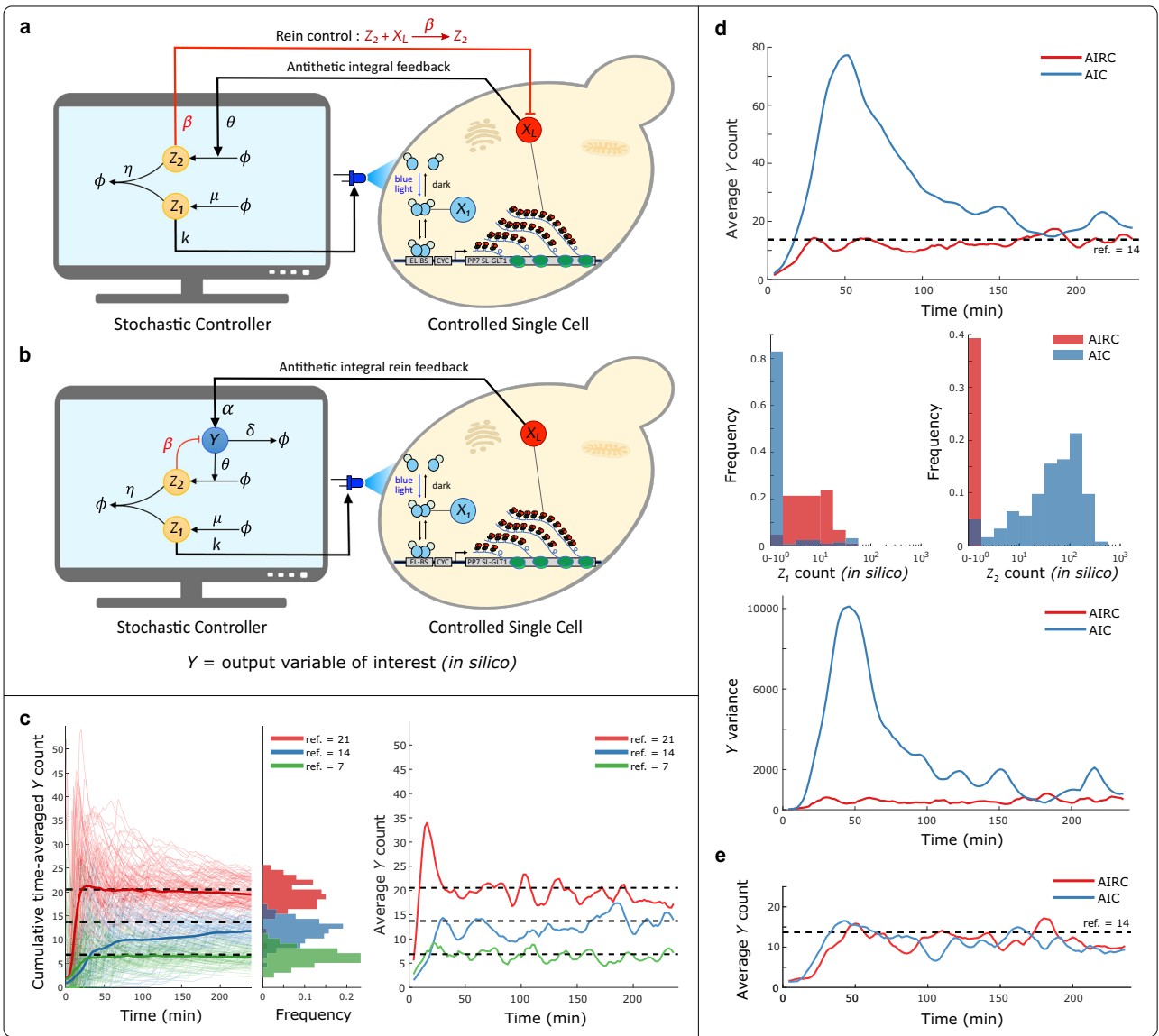

**Fig. 4 Antithetic Integral Rein Controller (AIRC) in Cyberloop. a** Addition of an extra reaction to the Antithetic Integral Controller (AIC) motif that incorporates a direct negative feedback from the controller species $Z_2$ to the system output $X_L$[40]. **b** Modification of the Cyberloop *in silico* setup to facilitate extra negative feedback from controller to the output. An extra molecule $Y$ (*in silico*) is added to the biological system network being controlled, and is considered as the new output variable of interest. This facilitates addition of an extra negative feedback reaction involving the output variable. **c** Demonstration of set-point tracking by the Antithetic Integral Rein Controller motif. Left: three Cyberloop runs with different reference values (dashed lines) were performed. Thin lines indicate cumulative time averages of $Y$ counts in individual cells, while thick lines represent the population average. Center: distribution of the average $Y$ count per cell over the course of the experiment. Right: population average of $Y$ count across cells as time progresses in those three experiments (Experimental parameters: $k = 0.1\,\text{min}^{-1}$, $\eta = 5\,\text{min}^{-1}$, $\theta = 0.02\,\text{min}^{-1}$ and $\mu = (7, 14, 21) \times \theta\,\text{min}^{-1}$, $\alpha = 0.5\,\text{min}^{-1}$, $\delta = 0.05\,\text{min}^{-1}$ and $\beta = 5\,\text{min}^{-1}$; Number of cells: red - 100, blue - 89, green - 73). **d** Two experiments, one with AIC ($\beta = 0$) motif and another with AIRC ($\beta \neq 0$) motif having same values for the other parameters, were carried out. Top: shows the time-course evolution of average output $Y$ abundance. Center: shows steady-state (from 180 to 240 min) distribution of $Z_1$ and $Z_2$ copy-numbers over all the cells. Bottom: shows the output variance as a function of time in both experiments. In these experiments, AIRC clearly displays better transient dynamics with faster settling time and lower variance compared to AIC. Furthermore, AIRC requires very low abundance of controller species $Z_2$ to achieve set-point tracking as compared to AIC in these experiments (Experimental parameters: $k = 0.1\,\text{min}^{-1}$, $\eta = 5\,\text{min}^{-1}$, $\theta = 0.02\,\text{min}^{-1}$ and $\mu = 14 \times \theta\,\text{min}^{-1}$, $\alpha = 0.5\,\text{min}^{-1}$, $\delta = 0.05\,\text{min}^{-1}$ and $\beta = (0, 5)\,\text{min}^{-1}$; Number of cells: red - 89, blue - 122). **e** Two experiments with high degradation rate ($\delta = 0.5\,\text{min}^{-1}$) of output $Y$ molecule, one with AIC ($\beta = 0$) motif and another with AIRC ($\beta \neq 0$) motif having same values for the other parameters, were carried out. This plot shows the time-course evolution of average output $Y$ abundance in the two experiments. Here, AIRC displays no significant improvement in the transient dynamics compared to AIC when there is large enough output degradation rate $\delta$ (Experimental parameters: $k = 0.01\,\text{min}^{-1}$, $\eta = 5\,\text{min}^{-1}$, $\theta = 0.02\,\text{min}^{-1}$ and $\mu = 14 \times \theta\,\text{min}^{-1}$, $\alpha = 0.5\,\text{min}^{-1}$, $\delta = 0.5\,\text{min}^{-1}$ and $\beta = (0, 0.2)\,\text{min}^{-1}$; Number of cells: red - 54, blue - 62). Source data are provided as a Source Data file.

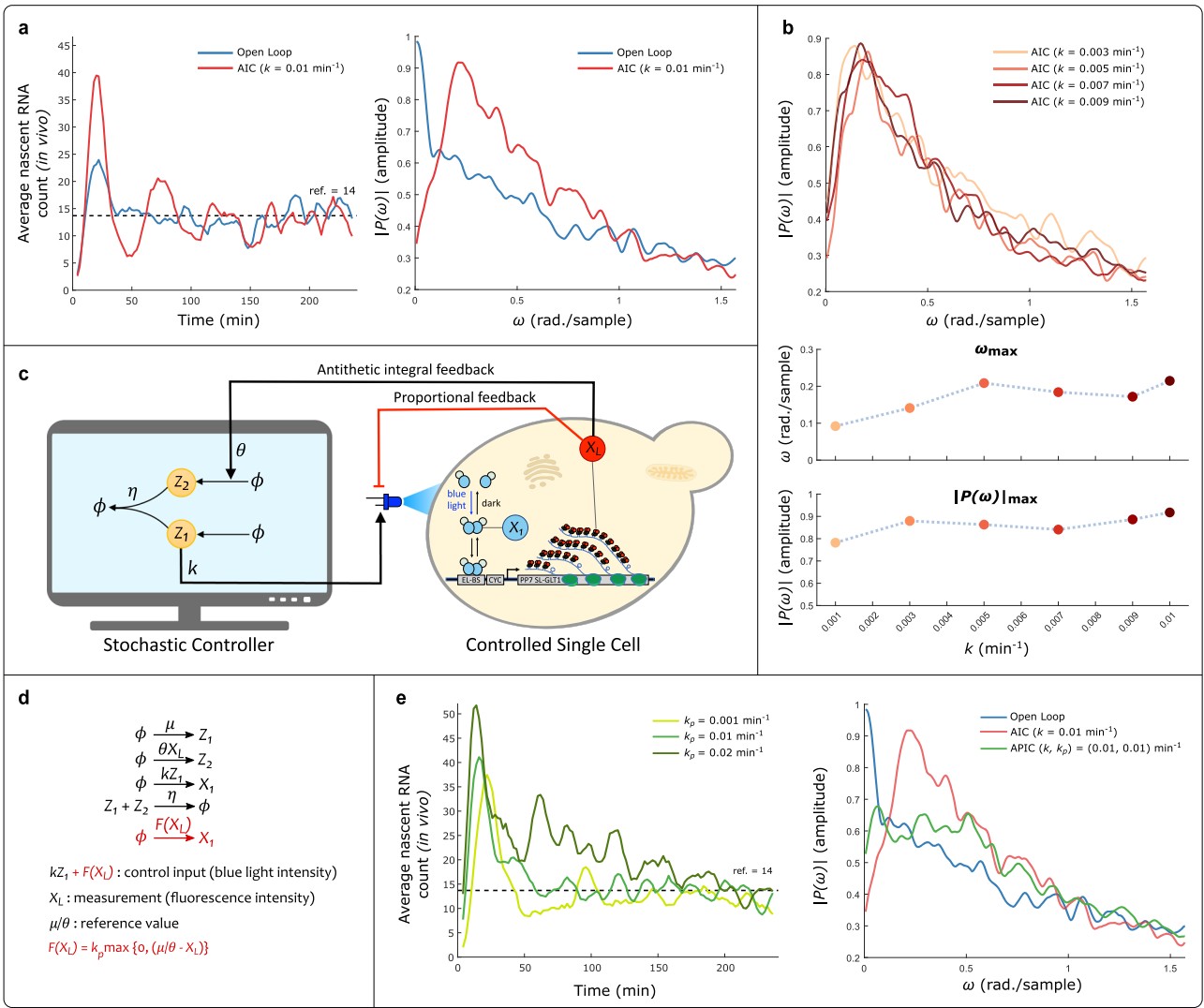

**Fig. 5 Single-cell trajectory investigation of Antithetic Integral Controller (AIC) and Antithetic Proportional-Integral Controller (APIC). a** These plots show results from two separate Cyberloop experiments, one with open-loop and another with closed-loop control (AIC). Left: shows the time-course evolution of average response (nascent RNA counts) of all the cells in the two experiments (these lines are filtered with five time-point averaging). Right: shows the average frequency response (amplitude spectrum) of all single-cell trajectories in these two experiments. For the closed-loop control case, a well defined peak in the average spectrum at a non-zero frequency indicates oscillatory single-cell output response. This is not observed in the open-loop average amplitude spectrum (Experimental parameters: $k = 0.01 \text{ min}^{-1}$, $\eta = 5 \text{ min}^{-1}$, $\theta = 0.2 \text{ min}^{-1}$ and $\mu = 14 \times \theta \text{ min}^{-1}$; Number of cells: red - 70, blue - 77). **b** Closed-loop control (AIC) experiments with different $k$ values. Top: shows the average frequency response in four separate experiments with the inset mentioned values of parameter $k$. Bottom: shows peak values $|P(\omega)|_{max}$ (bottom panel) in the amplitude spectrum and the frequency $\omega_{max}$ (center panel) corresponding to these peaks in these experiments. For all these closed-loop control experiments, the frequency response and associated attributes (peak and corresponding frequency) do not differ drastically with different $k$ values (Experimental parameters: $\eta = 5 \text{ min}^{-1}$, $\theta = 0.2 \text{ min}^{-1}$ and $\mu = 14 \times \theta \text{ min}^{-1}$; Number of cells: increasing value of parameter $k$ - 95, 51, 60, 69, 64, 70). **c, d** Antithetic Proportional-Integral Controller (APIC) in Cyberloop - addition of a proportional negative feedback action[41] to the Antithetic Integral Control motif. **e** Left: time-course evolution of average nascent RNA counts (system output) in three experiments with differing proportional gain $k_p$ values. Higher values of $k_p$ resulted in larger overshoots and longer settling times. Right: shows the average amplitude spectrum of single-cell trajectories in three experiments (open-loop, with AIC, and with APIC). With a suitable proportional gain value, the characteristic non-zero frequency response peak, observed in AIC case, is quenched in APIC case. This indicates that the extra proportional negative feedback action helps in mitigating oscillations in the output response (Experimental parameters: $k = 0.01 \text{ min}^{-1}$, $\eta = 5 \text{ min}^{-1}$, $\theta = 0.2 \text{ min}^{-1}$ and $\mu = 14 \times \theta \text{ min}^{-1}$; Number of cells: left plot with increasing value of parameter $k_p$ - 83, 86, 76, and right plot open loop - 77, AIC - 70, APIC - 86). Source data are provided as a Source Data file.

**Antithetic proportional-integral (PI) controller: single-cell trajectories.** As mentioned previously, the Antithetic Integral Control motif has been shown to exhibit robust perfect adaptation characteristics in the population (mean) behavior of the target-controlled network[20]. Applications requiring well-behaved single-cell responses prompt a further investigation on the effect this motif has on single-cell dynamics. A careful stochastic

simulation study with simple gene-expression system model in[53] reveals that in certain parameter regimes single-cell output responses can follow oscillatory trajectories even when the population mean output is converging and stable.

To probe the single-cell level effect of this motif over a real biological system network, we analyzed the single-cell trajectory data obtained from different Cyberloop experiments. We

**Table 1 Plasmids.**

| Plasmid | Backbone | Insert | Source |
|---------|----------|--------|--------|
| pDB58 | pKERG105 | *ACT1*pr-VPEL222-*CYC1*term | 62 |
| pDB96 | pDZ306 | *GLT1*-5xELbs-*CYC180*pr-24xPP7SL | 33 |
| pDB97 | pRG205 | *MET25*pr-tdPCP-NLS-tdmRuby3-*CYC1*term | 33 |

*pr* promoter, *term* terminator.

employed spectrum analysis tools[54], which are extensively used in signal processing field to extract frequency response of a given signal (single-cell output trajectory in our case). First, we analyzed the frequency response in open loop and closed loop (Antithetic Integral Control) experiments (Fig. 5a, left). As seen in the average frequency response (amplitude spectrum) of all single-cell trajectories (Fig. 5a, right), a well-defined peak at a non-zero frequency is observed in the closed-loop spectrum. This suggests that for the given experimental parameters in the Antithetic control motif, single-cell trajectories exhibit fluctuations with their amplitude spectrum achieving a maximum at about 0.2 rad./sample frequency. Further, we observed that the frequency response characteristics (peak value and the corresponding frequency) remains relatively unchanged for a range of controller gain parameter, $k$, values (Fig. 5b). Please refer to Methods section for further details about how the average frequency response of single-cell trajectories was computed in this study.

Next, we investigated the effect of adding a proportional feedback on the performance of the Antithetic Integral Control motif. For our experiments, we included a simple proportional negative feedback (Fig. 5c, d) with a max function which restricts the propensity function to non-negative values. This additional feedback term can represent a local approximation of an implementable repressing reaction[41]. As expected, this Antithetic Proportional-Integral Control motif shows average output set-point tracking and maintains robust perfect adaptation property (Fig. 5e, left). Here, we also observe that a higher proportional gain, $k_p$, value results in larger overshoot and longer settling times in the average output response[41]. Remarkably, this additional proportional negative feedback is found to mitigate oscillations observed in single-cell trajectories when employing the Antithetic Integral Control with the given parameters (Fig. 5e, right). This demonstrates a possible strategy to avoid oscillatory single-cell dynamics, which can be immensely useful in the biological implementation of integral controllers.

## Discussion

In this work we introduced the Cyberloop, an experimental framework for the rapid testing and optimization of biomolecular controllers embedded in the cellular environment. Using this tool, we first studied how a controller designed to be used in deterministic settings, the Autocatalytic Integral Controller, performs when implemented as a stochastic chemical reaction network. Through this example, we could observe the large discrepancies in controller behavior between the deterministic and the stochastic implementations of the system; the deterministic variant shows perfect tracking and a knob to tune its metabolic load on the cell[21], whereas its stochastic counterpart quickly becomes trapped in an absorbing state, ceasing to be effective. Capitalizing on the rapid design-cycle iterations enabled by implementing the biomolecular controller network in a computer, we could create a set of guidelines to overcome the challenges imposed by stochasticity, as well as parameter regimes that show a good performance. Addition of a low basal expression of the controller species, $V$, eliminated the absorbing state and led to an improved performance.

In another example, we analyzed the Antithetic control motif and how its performance is influenced by its constituting parameters. In contrast to the Autocatalytic controller, the Antithetic motif had previously been shown to benefit from the stochastic fluctuations and exhibit robust perfect adaptation in stochastic settings[20]. In our experiments with this motif, we indeed observed robust set-point tracking and demonstrated various ways of improving the performance (output settling time, overshoot, steady-state behavior, and controller species abundance). Most notably, we investigated a key assumption required for this controller to achieve perfect tracking: the lack of degradation reactions on the controller species[28,37]. This assumption is hard to achieve, particularly in fast dividing cells, where dilution will result in an 'effective' degradation of the controller species. In this regard, we analyzed the effect of dilution rates comparable to yeast's growth rate on the controller performance. We observed that although, according to theoretical analysis, dilution gives rise to steady-state error (Supplementary Information Section 1.2.1), it has a fairly negligible impact on the tracking performance for the target controlled biological process in our experiments, and hence may not warrant an increased complexity of biomolecular schemes that seek to minimize dilution-induced tracking errors[28,37]. Furthermore, as in the case of the Autocatalytic controller, we extract general guidelines to guide the design process, such as the need for a moderate to high values of the sequestration reaction rate, and a good performance of the controller with very low abundances of the controller species. We could also validate the performance improvement (reduced settling time and variance) with an extension of this controller, Antithetic Integral Rein Controller motif, which had only been analyzed theoretically before[40].

Investigating the single-cell dynamics with frequency analysis tools, we demonstrated that for certain parameters, the Antithetic control motif can induce an oscillatory single-cell output response[53] in our target biological system. We further demonstrated a possible strategy in suppressing oscillations in the closed loop single-cell dynamics by adding an extra proportional negative feedback action. Although this additional proportional feedback action had been analyzed theoretically and shown to reduce the output variance with improved controlled performance[41], we have exhibited its potential in also improving the closed loop single-cell dynamics with dampened oscillations.

With these diverse examples, we have established that the Cyberloop can act as an intermediate step between a design process fully reliant on simulations, and trial-and-error of different biological implementations. We consider this stepping stone between the two approaches valuable, as it enables one to transition from the idealization of biomolecular circuits to their testing in short time-spans. In contrast to fully simulating the system, the Cyberloop needs to make no assumptions regarding the structure or parametrization of the controlled network, as it is embedded in its biological context. However, the required flexibility to specify biomolecular controller architecture and parametrization also implies that these choices are made through the subjective judgement of the experimenter. Because of this, throughout this study we explored the parameter space for production and degradation of controller species in order to gauge the controller's sensitivity to these variables. The controller species abundance was varied from a few molecules per cell to a few thousands, consistent with physiological observations of regulatory molecules in *S. cerevisiae*. In addition, rates of production and degradation of these species were kept at biologically meaningful ranges, going from few reactions taking place per minute, to a maximum of approximately a hundred reactions taking place each second.

**Table 2 S. cerevisiae strains.**

| Strain | Genotype | Source |
|--------|----------|--------|
| BY4741 | MAT**a** his3Δ1 leu2Δ0 met15Δ0 ura3Δ0 | Euroscarf |
| BY4742 | MATalpha his3Δ1 leu2Δ0 lys2Δ0 ura3Δ0 | Euroscarf |
| DBY41 | BY4741, *LEU2::ACT1*pr-VPEL222-*CYC1*term(pDB58) | [62] |
| DBY41 | BY4741, *LEU2::ACT1*pr-VPEL222-*CYC1*term(pDB58) | [62] |
| DBY80 | DBY41, *GLT1*prΔ::*HIS3*-5xELbs-*CYC180*pr-24xPP7SL(pDB96) | [33] |
| DBY91 | BY4742, *URA3::MET25*pr-tdPCP-NLS-tdmRuby3-*CYC1*term(pDB97) | [33] |
| DBY96 | DBY80 mated with DBY91 | [33] |

*pr* promoter, *term* terminator.

In conclusion, we would like to remark the potential use of the Cyberloop framework for many wider applications involving optimization and design of other categories of synthetic circuits, beyond biomolecular controller circuit design. The Cyberloop also enables the establishment of virtual cell-to-cell communication which in turn can be used to exhibit pattern formation[55] or to engineer emerging behaviors[56]. It may also prove useful in carrying out tasks requiring division of labor among several subpopulations in order to achieve complex goals[57,58].

## Methods

**Growth conditions**. Unless otherwise indicated, *Escherichia coli* were grown in 14 mL tubes (Greiner) in LB (1% tryptone, 0.5% yeast extract, 1% NaCl) and incubated in an environmental shaker (Excella E24, New Brunswick) at 37 °C with shaking at 230 rpm. *Saccharomyces cerevisiae* were grown in 14 mL tubes (Greiner) in SD dropout medium (2% glucose, low fluorescence yeast nitrogen base (ForMedium), 5 g/L ammonium sulfate, 8 mg/L methionine, pH 5.8) and incubated in an environmental shaker (Innova 42R, New Brunswick) at 30 °C with shaking at 230 rpm unless otherwise indicated. Ampicillin (Sigma-Aldrich Chemie GmbH) was used at a concentration of 100 μg/mL.

**Plasmid and yeast strain construction**. No new plasmids or new *S. cerevisiae* strains were constructed in this study. The yeast strain (Fig. 1b) used in this work has already been constructed and published in[33]. As mentioned in[33], *S. cerevisiae* strains were derived from BY4741 and BY4742 (Euroscarf, Germany). *E. coli* TOP10 cells (Invitrogen) were used for plasmid cloning and propagation. Plasmids were constructed by restriction-ligation cloning using enzymes from New England Biolabs (USA). All plasmids, strains and related details are summarized in Tables 1 and 2. Strain DBY96 was used for all the experiments in this study.

**Culture media and initialization**. Yeast cell cultures were started from a −80 °C glycerol stock at least 24 hours prior to the experiment, and were grown at 30 °C in SD dropout medium. They were kept at $OD_{60} < 0.2$ in an incubator at 30 °C for the last 12 hour leading to the experiment. For each experiment, ~400 μL of cell culture was centrifuged at 3000 RCF for 6 min, and then sufficient volume of supernatant was removed to get a concentrated culture with $OD_{600} \sim 4$.

**Agarose pad preparation**. Around 800 μL of 2% agarose (UltraPure™Agarose, Invitrogen) in SD medium solution was poured on a microscope slide set-up. The set-up consisted of two stacks of microscope slides, each stack having two slides, placed 15 mm apart parallel to each other on another microscope slide, creating a U-shaped flat bottom well. A 25 mm × 25 mm square cover slip was then gently placed on top. It was left at room temperature for one hour. Once the agarose solution solidified, the microscope slide stacks and cover slip were removed, and the pad was cut on the sides with a scalpel to get ~15 mm × 15 mm flat-top pad. 1 μL of concentrated cell culture (as described above) was pipetted at the center of the pad. This pad was then carefully turned over, avoiding air-pockets, into a circular tissue culture dish with cover glass bottom (35 mm FluoroDish™, World Precision Instruments) lined inside with a strip of damp paper towel to maintain humidity during the experiment. The dish was closed, sealed with a strip of parafilm, then immediately placed onto a custom built sample holder inside the microscope's environmental box (Life Imaging Services, Switzerland). Cells were allowed to settle for one hour before starting experiments. As cells proliferated in mono-layer under the agarose pad only up to 5–6 hours and our light projection hardware[33] can only target cells proliferating in 2D layer, the experiment duration was set to four hours.

**Imaging and light delivery system**. All images were taken under a Nikon Ti-Eclipse inverted microscope (Nikon Instruments), equipped with a 40X oil-immersion objective (MRH01401, Nikon AG, Egg, Switzerland) and CMOS camera

ORCA-Flash4.0 (Hamamatsu Photonic, Solothurn, Switzerland). Following imaging set-ups were used in the microscope:

- Brightfield imaging - LED 100 (Märzhäuser Wetzlar GmbH & Co. KG) with diffuser and green interference filter placed in the light path
- Fluorescence (mRuby3) imaging - Spectra X Light Engine fluorescence excitation light source (Lumencor, Beaverton, USA), 550/15 nm LED line from the light source, 561/4 nm excitation filter, HC-BS573 beam splitter, 605/40 nm emission filter (filters and beam splitter acquired from AHF Analysetechnik AG, Tubingen, Germany)

In all of the experiments in this study, imaging/sampling was done at an interval of 2 min and the total duration of experiment was four hours. At every sampling time, two brightfield images above and below the focal plane (±5 a.u. Nikon Perfect Focus System) were acquired, with an exposure of 100 ms each, for cell segmentation and tracking. Subsequently, five fluorescence images (Z stacks with step size ~ 0.5 μm) were also captured, with an exposure of 300 ms each, for quantification (nascent RNA count).

The microscope was placed inside an opaque environmental box (Life Imaging Services, Switzerland), which maintained the temperature inside at 30 °C and also shielded the cell sample from external light.

A DMD (Digital Micromirror Device) based custom-built blue light projection set-up, developed in[33], was used for single-cell optogenetic stimulation under the microscope. A neutral density filter (ND 1.3, 25 mm absorptive filter from Thorlabs) was placed in the light stimulation pathway to reduce the blue light intensity reaching the cells. An open-source microscope control software YouScope[59] was used to operate and control the microscope as well as the light projection system.

**Workflow between consecutive sampling**. In a Cyberloop experiment, the workflow between consecutive sampling involved the following automated steps/processes running on a computer system:

1. Capture brightfield and fluorescence images of the cells placed under the microscope (using YouScope[59] microscope control software).
2. Segment individual cells in the brightfield image using CellX[60] cell segmentation tool.
3. For each segmented cell:
   i. Assign the segmented cell to its respective unique tracking index/identifier[33,61] in order to track the cell and acquire time trajectory for that cell.
   ii. Using captured fluorescence images, quantify the cellular readout spot intensity[33], which is proportional to the nascent RNA count of the segmented cell.
   iii. Based on the cellular readout (nascent RNA count), run an stochastic simulation[51] of the biomolecular controller network for a sampling duration (two minutes), and compute the input blue light intensity for the segmented cell.
4. Once the input blue light intensities for all segmented cells have been computed, generate a corresponding input mask image for the light projection system[33].
5. Correct the generated input mask image for optical aberrations in the projection optics (using calibration data computed just before the experiment)[33].
6. Project the corrected input mask image onto the cells placed on the sample plane of the microscope such that each segmented cell receives its corresponding blue light intensity computed by its own independent controller simulation in Step 3.iii.

Respective software routines for the above mentioned steps were developed and run in MATLAB (MathWorks) environment.

**Stochastic simulation of biomolecular controllers**. All in silico simulations of biomolecular contollers were run in MATLAB (MathWorks) environment.

Gillespie's Stochastic Simulation Algorithm (SSA,[51]) was used to simulate a given biomolecular controller reaction network for every cell individually. At every sampling time $t_k$, once the cellular readouts (nascent RNA counts) are quantified from acquired fluorescence images, these computational steps were followed for each tracked cell:

1. The quantified cellular readout (nascent RNA count) was used to compute and update propensities of the biomolecular controller reaction network.
2. Gillespie's SSA was applied for the image sampling time interval (two minutes) to obtain the controller species abundance.
3. This abundance value was then used to determine the blue light intensity $I(t_k)$ the corresponding cell should receive in the next cycle.

In this way, the controller output (blue light intensity $I(t_k)$) was updated once a new measurement (microscopy images) became available, and was held constant between measurement times. Due to an upper power limit on the light intensity output from the DMD projector, the applied light intensity at time $t_k$ was $\min(I(t_k), 1)$, where one corresponds to the maximum (scaled) light intensity that DMD projector can provide[33].

**Mean and Steady-state calculations**. When selecting a biological system for our study, we sought to use a highly stochastic system that will allow us to explore the impact of our controller in highly noisy environments. Therefore our target biological system network (Fig. 1b) was centered around transcription which occurs in highly random bursts[33]. As a result the controlled biological molecule "nascent RNA" also displays significant bursting behavior and highly stochastic dynamics (Supplementary Fig. 5) as desired. For such systems it is the expected value of the signal that is guaranteed to track the reference under antithetic integral feedback control[20]. There are two ways one can check this: by looking at sample averages of multiple traces or (for an ergodic system) by looking at time averages. We chose the former, and that is why we use average attributes in our experimental results. Although we are plotting trace averages, we are still performing control at the single-cell level, which gives dramatically better performance than an alternative scheme that uses the population average for feedback control.

All experiments in this study are 4 hours long, and with a sampling/imaging period of 2 min we get 121 sample-points throughout the duration of the experiment. All time-course mean/average attributes, for example, average nascent RNA count (average output), etc., are computed taking average over all the cells (which were tracked/targeted during the full experimental duration) for each sample-points. Let us consider that a total of $N$ individual cells were tracked/targeted in an experiment. Given the output time trajectory for cell $n$ as $\{s_{nk}, k = 1, 2, …, 121\}$, the average/mean trajectory is given by:

$$\text{Mean output trajectory}, \mathcal{M} = \left\{ \frac{\sum_{n=1}^{N} s_{nk}}{N}, k = 1, 2, …, 121 \right\}. \quad (1)$$

We further filter this mean trajectory with five time-point averaging filter. This filtered trajectory has been used to display mean attributes in all the figures, where applicable, presented in this study.

Based on the experimental data, we consider all samples from sample-point 30 (60 min) to sample-point 101 (202 min), if not mentioned otherwise, for the calculation of steady-state attributes such as mean steady-state error. We use the following steady-state calculations in our analysis:

$$\text{Mean steady-state output value for } n^{th} \text{ cell}, m_n = \frac{\sum_{k=i}^{j} s_{nk}}{j - i + 1}, \quad (2)$$

$$\text{Mean steady-state output value for all cells}, M = \frac{\sum_{n=1}^{N} m_n}{N}, \quad (3)$$

$$\text{Mean steady-state output error}, E_r = M - ref, \quad (4)$$

$$\text{Percentage mean steady-state output error}, E_r\% = \frac{M - ref}{ref} \times 100, \quad (5)$$

where $(i, j) = (30, 101)$ defines the steady-state time points, and $ref$ represents the set-point reference value.

For the steady-state distribution of controller species abundance (for example, Fig. 3f, right), we first compute the mean controller species count for each cell at steady state sample-points, and then we use these respective mean counts for all the tracked/targeted cells to get the required distribution.

**Frequency response computation (MATLAB)**. Given, single-cell output (nascent RNA count) trajectories $\{\mathcal{S}_n, n = 1, …, N\}$ at steady-state corresponding to $N$ cells in an experiment

$$\mathcal{S}_n = \{s_{nk}, k = 1, 2, 3, … \}, \quad (6)$$

where $s_{nk}$ is the measured nascent RNA count of $n^{th}$ cell at time $t_k$.

For $n = 1, …, N$, the corresponding Discrete Fourier Transform (DFT) of each trajectory can be computed as

$$\text{DFT}_n = \text{fft}(\mathcal{S}_n), \quad (7)$$

where fft($\cdot$) is an inbuilt MATLAB function which computes the DFT of a given signal. This $\text{DFT}_n$ is the frequency response corresponding to $n^{th}$ cell output trajectory.

Next, the average frequency response of all single-cell trajectories in an experiment can be computed by simply taking the mean of the individual DFTs of all tracked cells.

$$\text{Average frequency spectrum} = \text{mean}(\text{DFT}_n, n = 1, …, N) \quad (8)$$

**Results – analysis and formatting**. All the results/data in this study were analyzed and plotted using MATLAB R2018a (academic use) platform. These MATLAB generated plots/graphs were finally structured and formatted together as different figures of this article using Inkscape (v0.92, open source).

**Reporting summary**. Further information on research design is available in the Nature Research Reporting Summary linked to this article.

## Data availability

Raw data (MATLAB *.mat* files) for all results/figures presented in this article, including Supplementary figures, are available in the Source Data file. Any additional data are available upon request. Source data are provided with this paper.

## Code availability

The custom code used in this study is run on an integrated experimental set-up and hardware[33], and can not be executed without the full associated hardware-software suite. Code and hardware configuration files are available upon request.

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

## Acknowledgements

The authors would like to thank Dr. Dirk Benzinger for providing yeast strains, Fabian Rudolf (ETH Zurich) for the cell tracking code, and Dr. Stephanie Aoki for proof reading the manuscript. This project has received funding from the European Research Council (ERC) under the European Union's Horizon 2020 research and innovation programme (CyberGenetics; grant agreement no. 743269), and from a FET-Open research and innovation actions grant under the European Union's Horizon 2020 research and innovation programme (CyGenTiG; grant agreement no. 801041).

## Author contributions

M.K. conceived the project. S.K and M.R. developed the automation and control soft-ware, and performed the experiments. S.K. and M.R. analyzed the data. M.K. supervised the project and secured funding. S.K., M.R. and M.K wrote the manuscript.

## Competing interests

The authors declare no competing interests.
