## [Peer Review File · Nature Communications]

Reviewers' Comments:

Reviewer #1:

Remarks to the Author:

The manuscript by Kumar and colleagues presents an elegant "Hardware in the (cyber)loop" approach to testing candidate genetic controllers before they are implemented in vivo/in vitro. The authors claim that the approach they document allows them to extract general(isable) insights into the design/robustness analysis of in vivo controllers. The breadth and significance of the results the authors go on to present does not only suggest that this claim is credible, but also that these findings are truly novel and, crucially, actionable. There is no doubt in my mind that members of the community at the boundary between control engineering and synthetic biology will be able to take these results and translate them into genetic circuit designs that can attain robust control of target proteins. In light of these elements, I consider the manuscript by Kumar and colleagues an invaluable addition to our "gene circuit design/prototyping" toolbox; the only way to make it even more appealing would have been for the authors to "take their own advice" and show that such "actionable", yet "in silico", insights do actually translate into "in vivo/in vitro practice".

I only have few minor comments connected to the presentation and aspects that could benefit from clarifications:

- I found the description of how the HIL in-silico procedures work somewhat unclear (with specific ref to lines 110 and following): what *exactly* happens on the computer implementing the control algorithms after an image is taken? How long after the micrograph is taken is the input applied? Is it 1 sampling time (2 min)? How does this affect control performance (i.e. do the authors have, from simulation, an estimate of how this delay affects convergence/stability)? Connected: why have the authors chosen 2 min as sampling time? How does this depend on the characteristic time constant of the biological process they are observing/controlling? And, along the same lines: control experiments in yeast reported by other groups extend well below the 200-min mark in the experiments reported in this manuscript. Is there any specific aspect limiting the duration of these experiments?

- General comment on experimental data: it is not always clear from captions/text what is a *simulation* and what is data from in vivo experiments. I would encourage the authors to make explicit this distinction whenever confusion can arise -and given the quality of their results, this happens very often!

- When arguments around the robustness of control(lers') performance with respect to physiological parameters are made (e.g. cell cycle, e.g. Fig. 3), the authors should consider using Monte Carlo simulations drawing from a distribution of relevant parameters -rather than single point estimates. This would strengthen their arguments that would otherwise be relying on the the reader believing that, for example, all yeast cells have always, invariably, independently of media etc, a cell cycle duration of 80 min.

- regarding the AIRC configuration: it would be helpful to (a) report distributions of Z_1 and Z_2 as done in Fig. 2D and (b) report multiple set-point experiments at different levels. Also, after reading the section on the Rein Controller, I was left with the impression that the "lessons learnt" in this case were less striking. It would be beneficial to elaborate a little further on the actual significance of the authors' findings in this specific context.

Reviewer #2:

Remarks to the Author:

This paper presents Cyberloop, a hybrid experimental and simulation framework for rapid testing and refinement of biological controllers in single cells and populations of cells.

The tool is illustrated with two primary examples, the Autocatalytic integral controller and the Antithetic Controller (and the Antithetic Controller with REIN feedback). The key advance is to couple biomolecular stimulation or perturbation to optical light stimulation, generated as the

output of a controller simulation in silico. This allows an experimentalist to rapidly test many different control designs at the incremental cost of running fluorescence microscopy experiments of an "open-loop" or genetically unmodified system. As far as I can tell, the framework also can be used at any stage in biological design, e.g., if a single promoter has been incorporated into the genome, after first pass integration of a new landing site on the genome, or even after genomic integration of a complex circuit with multiple working components. At any of these stages, it is possible to ask how biomolecular control, effectuated by a proxy of optical light stimulation, would affect the dynamics of the system of interest.

The paper is a strong contribution to work at the interface of synthetic, systems biology, and control design. I recommend this paper for publication.

The primary contributions of this paper are an illustration of two class of controllers on fluorescently-tagged RNA reporters in yeast cells. Normally the study of a single class of controllers would be the subject of an entire paper, but with CyberLoop, they are able to rapidly implement both controllers by simply modulating the in silico model.

The work is quite convincing; I only have a few suggestions:

1) the reference setpoints for each of the controllers was formulated as a steady-state tracking problem or error from a static reference signal. While this is a fair illustration of classical control, controlled biological dynamics in nature often have dynamic setpoints, e.g., a limit cycle or a pulse train. I would be thrilled to see if the controller could be used to shape the plant response profile to match a square wave, for example. This would show that the controllers studied aren't one-sided or limited by diffusion/degradation rates.

2) I was a little confused as to why all the plots show averages of single cell traces as a measure of performance, when the preceding discussion centered on analysis of single cell control systems. It seems like the measure of effective single cell control is whether the output of a single cell adheres to a reference curve, not whether the average of many cell traces does. This can be misleading, especially if the mean of the traces is close to the reference, coincidentally, but the individual traces have a large spread. Finally, how many cell traces were averaged? Again, I think it would be better to see single cell traces and compare the performance of individual cells against the reference curve.

3) I would like to see more discussion that explores how CyberLoop can triage or differentiate control strategies that would be "nice to have" but are not feasible versus control strategies which have a correspondence with existing biological motifs. One of the biggest challenges in synthetic biology is that simulated biological networks don't behave the way they are expected to when fully cloned into genetic systems. I would expect there would be a significant deviation from the output of the CyberLoop control device and a designed realization of either of the controllers in vivo. If it is possible to realize one of these genetic circuits and show the performance when actually implemented in vivo is comparable, this would be a tremendous contribution. However, that may be beyond the scope of this paper, so I only offer it as a soft suggestion, not as a requirement for publication.

Overall, the Cyberloop system is the first software for implementing single cell control with hardware in the loop for in vivo systems. The authors study the effect of absorbing states in stochastic control strategies and show how to mitigate some of the challenges in dealing with single cell fluctuations. It is a first of a kind work that I think, with some minor revision, deserves to be published.

Response to Reviewers

Rapid Prototyping and Design of Cybergenetic Single-Cell Controllers

Sant Kumar, Mark Rullan, Mustafa Khammash

We are grateful to the editor for evaluating our work for further consideration, and to the two reviewers for their assessment of our work and for providing us with their valuable feedback. The suggestions from the reviewers have strengthened our manuscript and improved its clarity.

Based on the reviewers' comments, the following modifications have been incorporated in the revised manuscript. These changes have been highlighted in **red** in the manuscript text.

1. We modified the abstract to adhere to the word count limit requirement (150 or fewer).
2. We added figure 4S (supplement), briefly explaining the *in silico* steps of the Cyberloop framework. We also expanded Results (The Cyberloop) section in the manuscript, briefly discussing the sampling time period and actuation delay in the Cyberloop framework.
3. We added figure 8S (supplement), showing the computation times of running our software routine for samples/images with different number of segmented cells.
4. We added details about the number of cells targeted/tracked in every experiment (figure captions).
5. We added new dynamic set-point tracking experimental results (figure 6S, supplement).
6. We added new experimental results (in Antithetic Integral Controller section and figure 3g) that explore the effect of dilution on controller performance, with the doubling rate of cells being randomly sampled from a relevant distribution (figure 7S, supplement). In the previous manuscript version, we had considered doubling time of all cells to be the same in these experiments.
7. We added new experimental results in Antithetic Integral Rein Controller section and figure 4c,d,e showing multiple set-point tracking experiments, steady-state distributions of Z_1 and Z_2 , and experiments with higher output degradation rate. We also added figure 9S (supplement) displaying results from experiments having a basal production reaction of the output molecule.
8. We added figure 5S (supplement) showing some single cell traces and averaged trajectory from one Cyberloop experiment. We also added figure 3d in the manuscript showing three single cell traces from an experiment.
9. We added mean trajectory computation and related details in the Methods section in the manuscript.

Below are the two reviewer's comments and our point-by-point response (highlighted in **blue**).

Reviewer - 1

The manuscript by Kumar and colleagues presents an elegant "Hardware in the (cyber)loop" approach to testing candidate genetic controllers before they are implemented *in vivo/in vitro*. The authors claim that the approach they document allows them to extract general(isable) insights into the design/robustness analysis of *in vivo* controllers. The breadth and significance of the results the authors go on to present does not only suggest that this claim is credible, but also that these findings are truly novel and, crucially, actionable. There is no doubt in my mind that members of the community at the boundary between control engineering and synthetic biology will be able to take these results and translate them into genetic circuit designs that can attain robust control of target proteins. In light of these elements, I consider

the manuscript by Kumar and colleagues an invaluable addition to our “gene circuit design/prototyping” toolbox; the only way to make it even more appealing would have been for the authors to “take their own advice” and show that such “actionable”, yet “*in silico*”, insights do actually translate into “*in vivo/in vitro* practice”. I only have few minor comments connected to the presentation and aspects that could benefit from clarifications.

We thank the reviewer for her/his comments and appreciation of our work, and for their support for the publication of our manuscript.

1. I found the description of how the HIL *in-silico* procedures work somewhat unclear (with specific ref to lines 110 and following): what *exactly* happens on the computer implementing the control algorithms after an image is taken? How long after the micrograph is taken is the input applied? Is it 1 sampling time (2 min)? How does this affect control performance (i.e. do the authors have, from simulation, an estimate of how this delay affects convergence/stability)? Connected: why have the authors chosen 2 min as sampling time? How does this depend on the characteristic time constant of the biological process they are observing/controlling? And, along the same lines: control experiments in yeast reported by other groups extend well below the 200-min mark in the experiments reported in this manuscript. Is there any specific aspect limiting the duration of these experiments?

This is a very useful remark. We have now included an additional figure in the supplement explaining in more detail the *in silico* framework of the Cyberloop. The same figure is appended below to answer the reviewer’s questions.

As mentioned in the above figure, the input is applied after a delay of 2 minutes which is one sampling time period. As shown in the previous study (Figure 4C in [1]), for the biological target system (Figure 1b in the manuscript) that we have used in our experiments, this small delay has negligible impact on the tracking properties of controllers in the Cyberloop. Also, the system response time (the time at which half-maximal average nascent RNA counts are reached against a step light input stimulation) has been found to be at least 7.8 minutes (Figure 2A in [1]) which is considerably longer than the sampling time period or input actuation delay of 2 minutes. Hence, this fast sampling and actuation time (compared to the system response time) is expected to not induce any noticeable convergence/stability issues in the closed loop control.

The limiting factor in deciding the sampling time in our experiments has been the computation time needed to run cell segmentation, cell tracking, quantification and stochastic controller simulation routines. In our experiments, we have used the software tools developed in [1] (based on [2] and [3]) for segmentation, tracking and quantification of individual cells. Our software framework is sequential and thus the computation time is directly proportional to the number of cells segmented in the microscopy image, as seen in the above figure (showing 2057 data points). If needed, a shorter sampling time can be achieved by using parallel processing for segmentation, tracking, and quantification.

The duration of all of the experiments presented in this study is 240 minutes (4 hours). The limiting factor in this case is the microscopy cell-culture set-up that we use for the Cyberloop experiments under the microscope. As described in the Methods section in the manuscript, we use an Agarose pad for the cells to proliferate in mono-layer under the microscope. Currently, our light projection hardware [1] only allows us to target cells proliferating in 2D layer. The agarose pad lets the cell population stay in mono-layer only until 5-6 hours. After that, as the cell density increases, cells start growing in multiple layers as there is no rigid confinement. This renders our light projection hardware less effective in targeting individual cells accurately. This is the reason we limit our experiment duration to 4 hours. One way to get around this limitation is by using micro-fluidic devices which can provide more stable confined environment restricting the cells to proliferate in mono-layer for a longer duration. We are currently pursuing this technology to make it work on our Cyberloop setup reliably.

2. General comment on experimental data: it is not always clear from captions/text what is a *simulation* and what is data from in vivo experiments. I would encourage the authors to make explicit this distinction whenever confusion can arise -and given the quality of their results, this happens very often!

This is a very good observation—thanks for bringing it to our attention. We have now added “*in silico*” or “*in vivo*” remark, where applicable, in all the figure graph/plot labels in the manuscript.

3. When arguments around the robustness of control(lers’) performance with respect to physiological parameters are made (e.g. cell cycle, e.g. Fig. 3), the authors should consider using Monte Carlo simulations drawing from a distribution of relevant parameters -rather than single point estimates.

This would strengthen their arguments that would otherwise be relying on the the reader believing that, for example, all yeast cells have always, invariably, independently of media etc, a cell cycle duration of 80 min.

We thank the reviewer for her/his notable suggestion. We have now replaced the corresponding experimental results in the manuscript with new experimental data incorporating this suggestion as shown in Figure 3g.

4. regarding the AIRC configuration: it would be helpful to (a) report distributions of Z_1 and Z_2 as done in Fig. 2D and (b) report multiple set-point experiments at different levels. Also, after reading the section on the Rein Controller, I was left with the impression that the "lessons learnt" in this case were less striking. It would be beneficial to elaborate a little further on the actual significance of the authors' findings in this specific context.

The steady-state distribution of Z_1 and Z_2 are now shown in Figure 4d, and multiple set-point experimental results have been added as Figure 4c in the manuscript. Furthermore, we have added some new observations and new results pertaining to the Antithetic Integral Rein Controller. Please review Figure 4 (manuscript), Figure 9S (supplement), and the second paragraph (highlighted in red) in the "Antithetic Integral Rein Controller" section in the manuscript.

Reviewer - 2

This paper presents Cyberloop, a hybrid experimental and simulation framework for rapid testing and refinement of biological controllers in single cells and populations of cells.

The tool is illustrated with two primary examples, the Autocatalytic integral controller and the Antithetic Controller (and the Antithetic Controller with REIN feedback). The key advance is to couple biomolecular stimulation or perturbation to optical light stimulation, generated as the output of a controller simulation *in silico*. This allows an experimentalist to rapidly test many different control designs at the incremental cost of running fluorescence microscopy experiments of an "open-loop" or genetically unmodified system. As far as I can tell, the framework also can be used at any stage in biological design, e.g., if a single promoter has been incorporated into the genome, after first pass integration of a new landing site on the genome, or even after genomic integration of a complex circuit with multiple working components. At any of these stages, it is possible to ask how biomolecular control, effectuated by a proxy of optical light stimulation, would affect the dynamics of the system of interest.

The paper is a strong contribution to work at the interface of synthetic, systems biology, and control design. I recommend this paper for publication.

The primary contributions of this paper are an illustration of two class of controllers on fluorescently-tagged RNA reporters in yeast cells. Normally the study of a single class of controllers would be the subject of an entire paper, but with Cyberloop, they are able to rapidly implement both controllers by simply modulating the *in silico* model. The work is quite convincing; I only have a few suggestions:

We thank the reviewer for his favorable assessment of our work and for recommending our manuscript for publication.

1. the reference setpoints for each of the controllers was formulated as a steady-state tracking problem or error from a static reference signal. While this is a fair illustration of classical control, controlled biological dynamics in nature often have dynamic setpoints, e.g., a limit cycle or a pulse train. I would be thrilled to see if the controller could be used to shape the plant response profile to match a square wave, for example. This would show that the controllers studied aren't one-sided or limited by diffusion/degradation rates.

The reviewer brings up an important pragmatic point. It is indeed the case that controlled biological dynamics in nature often exhibits dynamic set-point tracking. We have now added new experimental results in Figure 6S (supplement) demonstrating that a target biological network can achieve dynamic set-point tracking when coupled with Antithetic Integral Control motif. Here, we show up-shift and down-shift set-point changes in two separate experiments because we are limited to 240 minutes of experimental duration as discussed in the response to the first comment of reviewer 1.

- I was a little confused as to why all the plots show averages of single cell traces as a measure of performance, when the preceding discussion centered on analysis of single cell control systems. It seems like the measure of effective single cell control is whether the output of a single cell adheres to a reference curve, not whether the average of many cell traces does. This can be misleading, especially if the mean of the traces is close to the reference, coincidentally, but the individual traces have a large spread. Finally, how many cell traces were averaged? Again, I think it would be better to see single cell traces and compare the performance of individual cells against the reference curve.

When selecting a biological system for our study, we sought to use a highly stochastic system that will allow us to explore the impact of our controller in highly noisy environments. Therefore our target biological system network (used in this study) was centered around transcription (Figure 1b in the manuscript) which occurs in highly random bursts [1]. As a result the controlled biological molecule “nascent RNA” also displays significant bursting behaviour and highly stochastic dynamics (as desired). For such systems it is the *expected value* of the signal that is guaranteed to track the reference under antithetic integral feedback control. There are two ways one can check this: by looking at sample averages of multiple traces or (for an ergodic system) by looking at time averages. We chose the former, and that is why we plot averages. Although we are plotting trace averages, we are still performing control at the single-cell level, which gives dramatically better performance than an alternative scheme that uses the population average for feedback control. We have now added some single cell traces (from one of the Antithetic Control motif experiments) in figure 5S in the supplementary text and figure 3d in the manuscript to demonstrate what single cell traces look like. An example can be seen below. Single cell traces of *protein output* (not measured experimentally in this study) will be significantly smoother, as it would be “low-pass” filtered by the translation process.

As mentioned before, we have used average attributes to observe, characterize, and compare tracking performance of different controller motifs. However, to observe fluctuations and oscillations induced by the controller on a single-cell level we have considered a two step process: computing the frequency response of single cell traces first and then taking average of those individual frequency responses (instead of directly computing the frequency response of the averaged trajectory).

On an average 84 cells were tracked and targeted in each of the experiments presented in the manuscript. We have now added details about number of cells tracked in the experiment in all figures in the manuscript.

- I would like to see more discussion that explores how CyberLoop can triage or differentiate control strategies that would be “nice to have” but are not feasible versus control strategies which have a correspondence with existing biological motifs. One of the biggest challenges in synthetic biology is that simulated biological networks don’t behave the way they are expected to when fully cloned into genetic systems. I would expect there would be a significant deviation from the output of the CyberLoop control device and a designed realization of either of the controllers in vivo. If is it

possible to realize one of these genetic circuits and show the performance when actually implemented in vivo is comparable, this would be a tremendous contribution. However, that may be beyond the scope of this paper, so I only offer it as a soft suggestion, not as a requirement for publication.

This is indeed an excellent point. We consider this suggestion extremely important, and if realized successfully this could be a good contribution towards control design and synthetic biology disciplines. One way to go about it is to start with an existing genetically implemented control system whose performance has been characterized, and then to engineer the open loop to be optogenetically controllable so that the *in vivo* controller can be replaced by an *in silico* controller whose performance can be compared to that of the genetically engineered one. We are currently pursuing such an approach for a mammalian system in the lab. However, designing and engineering optogenetic actuators that are suitable for dynamic control in mammalian cells is a challenging task, and this work is still ongoing with results expected only 6 months to 1 year from now. Hence, as the reviewer noted, we believe that the suggested effort, while important and exciting, is beyond the scope of this manuscript.

Overall, the Cyberloop system is the first software for implementing single cell control with hardware in the loop for in vivo systems. The authors study the effect of absorbing states in stochastic control strategies and show how to mitigate some of the challenges in dealing with single cell fluctuations. It is a first of a kind work that I think, with some minor revision, deserves to be published.

We are very thankful to the reviewer for his supportive comments.

References

- [1] Marc Rullan, Dirk Benzinger, Gregor W. Schmidt, Andreas Miliadis-Argeitis, and Mustafa Khammash. An optogenetic platform for real-time, single-cell interrogation of stochastic transcriptional regulation. *Molecular Cell*, 70(4):745–756, 2018.
- [2] S. Dimopoulos, C. E. Mayer, F. Rudolf, and J. Stelling. Accurate cell segmentation in microscopy images using membrane patterns. *Bioinformatics*, 30(18):2644–2651, 2014.
- [3] M. Ricicova, M. Hamidi, A. Quiring, A. Niemistö, E. Emberly, and C. L. Hansen. Dissecting genealogy and cell cycle as sources of cell-to-cell variability in mapk signaling using high-throughput lineage tracking. *Proceedings of the National Academy of Sciences of the United States of America*, 110(28):11403–8, 2013.

Reviewers' Comments:

Reviewer #1:

Remarks to the Author:

The authors have addressed all the points I previously raised and, I am pleased to say, I am satisfied with the edits.

Reviewer #2:

Remarks to the Author:

The authors have fully addressed all my questions and points raised in my preceding review.

This is a great paper and I look forward to seeing the published version and sharing with colleagues.